# Outcomes of Robotic Pancreatectomy in the Octogenarian: A Multicenter Retrospective Cohort Study

**DOI:** 10.3390/cancers17183038

**Published:** 2025-09-17

**Authors:** Kosei Takagi, Yuichiro Uchida, Tomokazu Fuji, Takeshi Takahara, Kazuya Yasui, Takeyoshi Nishiyama, Ichiro Uyama, Koichi Suda, Toshiyoshi Fujiwara

**Affiliations:** 1Department of Gastroenterological Surgery, Graduate School of Medicine, Dentistry, and Pharmaceutical Sciences, Okayama University, Okayama 700-8558, Japan; pri958hs@s.okayama-u.ac.jp (T.F.); pjyv6nvp@s.okayama-u.ac.jp (K.Y.); me17063@s.okayama-u.ac.jp (T.N.); toshi_f@md.okayama-u.ac.jp (T.F.); 2Department of Surgery, Fujita Health University, Toyoake 470-1192, Japan; yuichiro.uchida@fujita-hu.ac.jp (Y.U.); takeshi.takahara@fujita-hu.ac.jp (T.T.);; 3Department of Advanced Laparoscopic and Robotic Surgery, Fujita Health University, Toyoake 470-1192, Japan; iuyama@fujita-hu.ac.jp

**Keywords:** robotic pancreatectomy, pancreatoduodenectomy, distal pancreatectomy, elderly patients

## Abstract

**Simple Summary:**

Contrary to the increasing incidence of pancreatic and periampullary cancers with advancing age, the evidence of robotic pancreatectomy for octogenarian patients is lacking. This study (n = 380) aimed to investigate the safety and feasibility of robotic pancreatectomy in octogenarian patients (≥80 years, n = 54). Octogenarian patients had more comorbidities and a higher incidence of malignant diseases, leading to significantly longer hospital, but had equivalent postoperative outcomes compared to younger patients (<80 years). Moreover, age (≥80 years) was not an independent risk factor for major complications following robotic pancreatectomy. This study suggested that robotic pancreatectomy can be a safe and feasible option for carefully selected octogenarian patients.

**Abstract:**

**Background/Objectives**: Due to the increasing incidence of pancreatic and periampullary cancers with advancing age, coupled with the growing evidence supporting minimally invasive pancreatectomy, the demand for such procedures is rising. However, data on the feasibility of robotic pancreatectomy in octogenarian patients remain scant. This study aimed to investigate overall outcomes of robotic pancreatectomy and evaluate its safety and feasibility in octogenarian patients. **Methods**: A multicenter, retrospective study was conducted, including 380 patients who underwent robotic pancreatectomy at two high-volume centers in Japan from April 2020 to December 2024. Using prospectively collected data, we compared outcomes between younger patients (<80 years) and octogenarian patients (≥80 years). Multivariable logistic regression analyses were performed to assess the impact of age on postoperative outcomes. **Results**: Among the 380 patients, with a median age of 72 (interquartile range: 61–77) years, 213 underwent robotic pancreatoduodenectomy (RPD), and 167 underwent robotic distal pancreatectomy (RDP). Octogenarian patients were found to have more comorbidities and a higher incidence of malignant diseases. Octogenarians experienced significantly longer hospital stays post-RPD (22 [octogenarian; n = 36] vs. 14 [younger; n = 177] days, *p* < 0.001) and post-RDP (14 [n = 23] vs. 10.5 [n = 144] days, *p* = 0.02), yet their perioperative outcomes were comparable. Multivariable analyses indicated that age (≥80 years) was not a significant risk factor for major complications following robotic pancreatectomy (odds ratio, 1.33; 95% confidence interval, 0.59–2.84; *p* = 0.479). **Conclusions**: This multicenter study conducted at high-volume centers suggests that robotic pancreatectomy can be safely performed in carefully selected octogenarian patients.

## 1. Introduction

The rapid pace of population aging presents a significant global challenge [1]. The rising incidence of pancreatic and periampullary cancers associated with aging suggests an increasing need for pancreatectomy among elderly patients [2]. Surgical decision-making for these patients must be approached with caution, particularly for those with complex comorbidities. However, advancements in surgical techniques and perioperative management have led to the viewpoint that age alone should no longer be seen as a contraindication for pancreatectomy [3].

Currently, expanding global evidence has supported minimally invasive pancreatectomy, especially robotic pancreatectomy [4,5]. While minimally invasive approaches generally offer better perioperative outcomes compared to open surgery [6], the appropriateness of robotic surgery for elderly patients remains a topic of debate. Previous research has affirmed the safety and feasibility of robotic pancreatoduodenectomy (RPD) in elderly cohorts (≥70 years or ≥75 years) [7,8]. However, its effectiveness and safety in octogenarian patients (≥80 years) are not well documented. Additionally, few studies have addressed the impact of patient age on the outcomes of robotic distal pancreatectomy (RDP).

This multicenter study aims to assess the perioperative outcomes of robotic pancreatectomy, including RPD and RDP. We also aim to evaluate the safety and feasibility of performing robotic pancreatectomy in octogenarian patients compared to younger cohorts.

## 2. Materials and Methods

### 2.1. Study Design

This multicenter, observational retrospective study included 380 consecutive patients who underwent robotic pancreatectomy at two high-volume centers, Fujita Health University Hospital (FHUH) and Okayama University Hospital (OUH), in Japan between April 2020 and December 2024. The study was approved by the Ethics Committees of our institutions (FHUH, approval no. CI24-444; OUH, approval no. 2407-023) and was conducted in accordance with the Declaration of Helsinki. Due to the study’s retrospective nature, the need for informed consent was waived.

### 2.2. Clinical Collection

Data were collected from prospectively maintained databases at the two institutions. Extracted data included sex, age, body mass index, American Society of Anesthesiologists (ASA) physical status, comorbidities (hypertension and diabetes), neoadjuvant chemotherapy, preoperative biliary drainage, primary disease (pancreatic cancer, bile duct cancer, intraductal papillary mucinous neoplasm, pancreatic neuroendocrine tumor, ampullary carcinoma, duodenal adenocarcinoma, and others, including benign tumors), type of procedure (RPD or RDP), operative time, blood loss, conversion to open surgery, pancreatic texture (soft or hard), diameter of the main pancreatic duct, length of postoperative hospital stay, major complications, 30-day and 90-day mortality, incisional and organ/space surgical site infections (SSI), postoperative pancreatic fistula (POPF), bile leakage, delayed gastric emptying (DGE), and postoperative pancreatitis. Patients were categorized into two groups based on age: younger patients (<80 years) and octogenarian patients (≥80 years).

### 2.3. Definition of Postoperative Complications

Conversion to open surgery was defined as an unplanned, urgent switch to open laparotomy. Postoperative complications occurring within 1 month after surgery were recorded and graded using the Clavien-Dindo classification, with major complications classified as grade ≥ 3 [9]. Definitions from the International Study Group on Pancreatic and Liver Surgery were employed to evaluate POPF [10], bile leakage [11], and DGE [12]. The clinically relevant POPF and DGE (≥grade B) were included.

### 2.4. Surgical Technique

Surgical protocols for RPD and RDP at FHUH are detailed in previous publications [13,14]. Similarly, protocols at OUH have been reported [15,16]. Both institutions serve as mentor centers for robotic pancreatectomy in Japan. Surgical protocols were standardized through mutual observation of each institution’s facilities.

The indications for robotic pancreatectomy were determined at multidisciplinary team meetings, with no age contraindications. The protocol included benign and malignant tumors that did not necessitate resection of other organs or vascular structures [15]. Currently, robotic pancreatectomy involving other organ and vascular resections is not covered by insurance in Japan.

### 2.5. Selection Criteria for Surgery

Patients meeting the following criteria were eligible for surgery: capable of making decisions and expressing their wishes regarding medical procedures and independent activity of daily living and mobility.

### 2.6. Statistical Analysis

Statistical analyses were conducted using JMP software v11 (SAS Institute, Cary, NC, USA). Data are presented as proportions for categorical variables and medians with interquartile ranges (IQR) for continuous variables. The Mann–Whitney U test was used for continuous variables and Fisher’s exact test for categorical variables. All reported *p*-values are two-tailed, with a significance threshold set at an alpha level of 0.05.

Initial analyses focused on patient characteristics and perioperative outcomes for RPD and RDP. Outcomes were compared across age groups. Subsequently, univariate and multivariable logistic regression analyses were performed to identify risk factors associated with major complications, with odds ratios (ORs) and 95% confidence intervals (CIs) calculated. All preoperative and operative variables with *p* < 0.01 in univariate analysis were included in multivariable analyses. Finally, postoperative survivals, including overall survival (OS) and recurrence-free survival (RFS), were investigated using the Kaplan–Meier method, and differences between curves were calculated using the log rank test.

## 3. Results

### 3.1. Implementation

The annual number of robotic pancreatectomies performed at the two institutions is shown in Figure 1. A gradual increase in robotic pancreatectomy was observed following its introduction in 2020.

Regarding pancreatoduodenectomy (PD) approaches over time, the proportion of robotic surgeries increased from 24.7% in 2020 to 66.7% in 2024 (Figure 2a). Conversely, laparoscopic distal pancreatectomy was gradually replaced by robotic surgery, with 76.8% of distal pancreatectomies being performed robotically in 2024 (Figure 2b).

### 3.2. Patient Demographics

This study included 380 patients, comprising 199 from FHUH and 181 from OUH. The baseline characteristics of the patients are shown in Table 1. The cohort consisted of 196 men and 184 women, with a median age of 72 years (IQR, 61–77). The most common primary diseases were pancreatic cancer (n = 115, 30.3%), intraductal papillary mucinous neoplasm or carcinoma (n = 83, 21.8%), and pancreatic neuroendocrine neoplasm (n = 40, 10.5%). The cohort included 213 cases of RPD and 167 cases of RDP.

When outcomes were stratified by age, octogenarian patients (≥80 years) were found to have higher ASA scores, more comorbidities, and a higher incidence of malignant diseases (Table 1).

### 3.3. Outcomes of Robotic Pancreatectomy

#### 3.3.1. Robotic Pancreatoduodenectomy

The overall short-term outcomes of 213 RPDs are summarized in Table 2. Preoperative biliary drainage was performed in 68 patients (31.9%). The median operative time was 490 min (IQR, 410–632), and the median blood loss was 100 mL (IQR, 38–242). The conversion rate to open surgery was 1.4%. Regarding postoperative outcomes, the 30-day and 90-day mortality rates were both 0.9%. The median postoperative hospital stay was 16 days (IQR, 11–23), with re-operation and major complication rates of 5.0% and 17.4%, respectively.

#### 3.3.2. Robotic Distal Pancreatectomy

Table 3 presents the overall outcomes of 167 RDPs. A spleen-preserving technique was used in 48 patients (28.7%), including the Kimura technique in 25 and the Warshaw technique in 23 patients. The median operative time was 281 min (IQR, 219–372), and the median blood loss was 50 mL (IQR, 9–110). No mortality was observed postoperatively. The median postoperative hospital stay was 11 days (IQR, 9–15), and the incidence of major complications was 7.8%.

### 3.4. Outcomes of Robotic Pancreatectomy Stratified by Age

#### 3.4.1. Robotic Pancreatoduodenectomy

Postoperative outcomes of RPD stratified by age are described in Table 2. No significant differences were observed in operative time or blood loss between the younger (<80 years) and octogenarian (≥80 years) groups. However, the octogenarian group had significantly longer hospital stays compared to the younger group (22 vs. 14 days, *p* < 0.001). No significant differences were observed in the rates of 30-/90-day mortality, major complications, incisional or organ/space SSI, POPF, or bile leakage. The only significant difference was a higher incidence of delayed gastric emptying (DGE) in the octogenarian group (*p* = 0.023).

#### 3.4.2. Robotic Distal Pancreatectomy

The comparison of RDP outcomes between the younger and octogenarian groups is shown in Table 3. No significant differences were observed in operative factors, including operative time and blood loss. However, postoperative hospital stays were significantly longer in the octogenarian group than in the younger group (14 [octogenarian] vs. 10.5 [younger] days, *p* = 0.023). Postoperative outcomes were comparable between the two groups. Although not significant, the octogenarian group tended to have more major complications than the younger group (17.4% vs. 6.3%, *p* = 0.097).

### 3.5. Risk Factors for Postoperative Major Complications After Robotic Pancreatectomy

The results of the univariate and multivariable analyses investigating risk factors associated with major complications following robotic pancreatectomy (50 events among 380 procedures) are presented in Table 4. Multivariable analysis showed that age (≥80 years) was not a significant risk factor for major complications after robotic pancreatectomy (OR, 1.33; 95% CI, 0.59–2.84; *p* = 0.479).

### 3.6. Long-Term Outcomes After Robotic Pancreatectomy

Following the median follow-up of 22.6 months (IQR, 12.7–36.3), 1-/2-/3-year OS and RFS were 97.4%/89.3%/86.9% and 87.9%/80.5%/75.7%, respectively. The octogenarian group had comparable OS to the younger group (*p* = 0.878; Figure 3a) but worse RFS (*p* = 0.003; Figure 3b). However, significant differences between the groups for RFS disappeared when the analysis was limited to patients with malignant diseases (Figure 3c,d).

## 4. Discussion

To the best of our knowledge, this is the first multicenter, retrospective study to investigate the overall outcomes of robotic pancreatectomy, including 213 RPDs and 167 RDPs, at two high-volume centers, FHUH and OUH, in Japan. The results demonstrated feasible outcomes for both RPD and RDP following the introduction of domestic insurance coverage in 2020. Additionally, this study assessed the impact of age on perioperative outcomes in patients undergoing robotic pancreatectomy. While octogenarian patients had longer postoperative hospital stays, their outcomes were comparable to those of younger patients following both RPD and RDP. Age was not found to be an independent risk factor for major postoperative complications after robotic pancreatectomy.

Regarding overall outcomes after RPD, our results were acceptable when compared to those reported by high-volume centers in the West and the international benchmark values for open PD [17,18,19]. On the other hand, the outcomes for RDP were relatively better than the benchmark values for RDP [20]. The introduction and dissemination of robotic pancreatectomy in Japan occurred later than in the West, beginning in 2020. However, our results suggest that we have successfully implemented the robotic surgery program in Japan.

A recent meta-analysis indicated the safety and feasibility of minimally invasive PD in elderly patients [21]. However, it showed that elderly patients had slightly higher rates of mortality, pulmonary complications, and major morbidity [21]. Moreover, data on the impact of age on outcomes following RPD remain limited. In the present study, we confirmed that RPD outcomes in octogenarian patients were comparable to those in younger patients, despite a significantly higher incidence of DGE and longer hospital stays in the octogenarian group. Furthermore, we found equivalent outcomes for RDP in octogenarian patients. Given the lack of research on RDP outcomes in elderly patients, our findings that RDP can be safely performed in octogenarians are significant and contribute to the growing body of evidence supporting its safety. However, a more cautious interpretation was advisable, acknowledging that the small sample size (n = 23) may be underpowered to detect a true difference in complication rates and that further study should be required. Preoperative characteristics in octogenarian patients, such as higher ASA scores, more comorbidities, and a higher incidence of malignant disease, may have contributed to the prolonged hospital stays observed after surgery. Accordingly, longer hospital stays in octogenarian patients may be explained by the fact that they required longer times for gastrointestinal and functional recovery.

Our multivariable analyses found that age (≥80 years) was not a risk factor for major complications following robotic pancreatectomy. While previous studies have demonstrated the safety of minimally invasive PD in elderly patients, the age cut-offs in those studies ranged from 65 to 75 years [21]. Therefore, this study represents the first investigation into the feasibility of robotic pancreatectomy in octogenarian patients. However, elderly patients generally have more age-related comorbidities, and their surgical risk is likely higher than that of younger patients [8,22,23]. Comprehensive perioperative management, including nutritional support and early mobilization, is crucial to facilitate postoperative recovery after surgery [24,25]. Careful patient selection is also particularly important for elderly patients undergoing robotic pancreatectomy.

This study has several limitations. Although it is a multicenter study, it is retrospective and involves a relatively small sample size. The introduction of potential selection bias in the patients who underwent robotic pancreatectomy cannot be denied. The huge differences in histology between younger and octogenarian patients could be a critical cofounder. Due to the retrospective and observational nature of the study, this study did not incorporate a sophisticated preoperative assessment of the patients’ functional or biological age into the baseline comparison. Assessment of frailty or sarcopenia may help to evaluate the patients’ fitness for surgery [26,27,28]. Additionally, outcomes following robotic pancreatectomy were not compared with those of open surgery. However, previous studies have reported comparable outcomes between RPD and open PD in elderly patients [29,30]. Moreover, this study investigated long-term outcomes in all cohorts and patients with malignant diseases; however, the results lacked disease-specific long-term outcomes. Since the indication for robotic pancreatectomy in octogenarian patients was a higher incidence of malignant diseases, our results may support the surgical and oncological adequacy of robotic surgery for octogenarian patients. Future studies should investigate the disease-specific long-term outcomes in octogenarian patients to provide further evidence on the role of robotic pancreatectomy in this population.

## 5. Conclusions

This multicenter study demonstrated acceptable overall outcomes for robotic pancreatectomy at two high-volume centers in Japan. Although octogenarians had higher ASA scores, more comorbidities, and a higher incidence of malignant diseases, robotic pancreatectomy appears to be a safe and feasible option for carefully selected octogenarian patients. Further investigation of long-term outcomes following robotic pancreatectomy should be required.

## Figures and Tables

**Figure 1 cancers-17-03038-f001:**
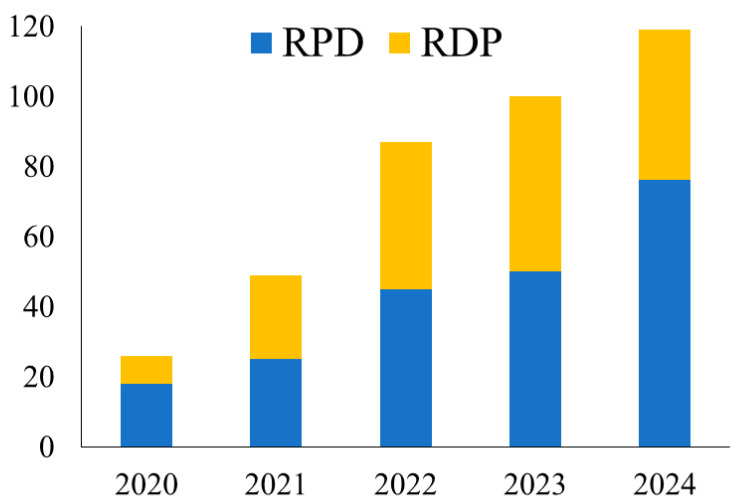
The annual total number of patients who underwent robotic pancreatectomy between April 2020 and December 2024 at two high-volume centers in Japan. RPD, robotic pancreatoduodenectomy; RDP, robotic distal pancreatectomy.

**Figure 2 cancers-17-03038-f002:**
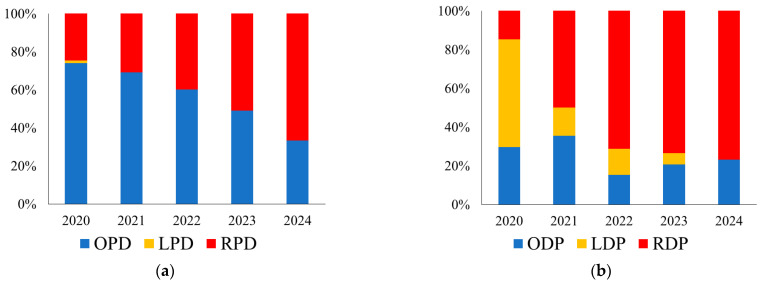
(**a**) Trends in surgical approaches to pancreatoduodenectomy over time. (**b**) Trends in surgical approaches to distal pancreatectomy over time. OPD, open pancreatoduodenectomy; LPD, laparoscopic pancreatoduodenectomy; RPD, robotic pancreatoduodenectomy; ODP, open distal pancreatectomy; LDP, laparoscopic distal pancreatectomy; RDP, robotic distal pancreatectomy.

**Figure 3 cancers-17-03038-f003:**
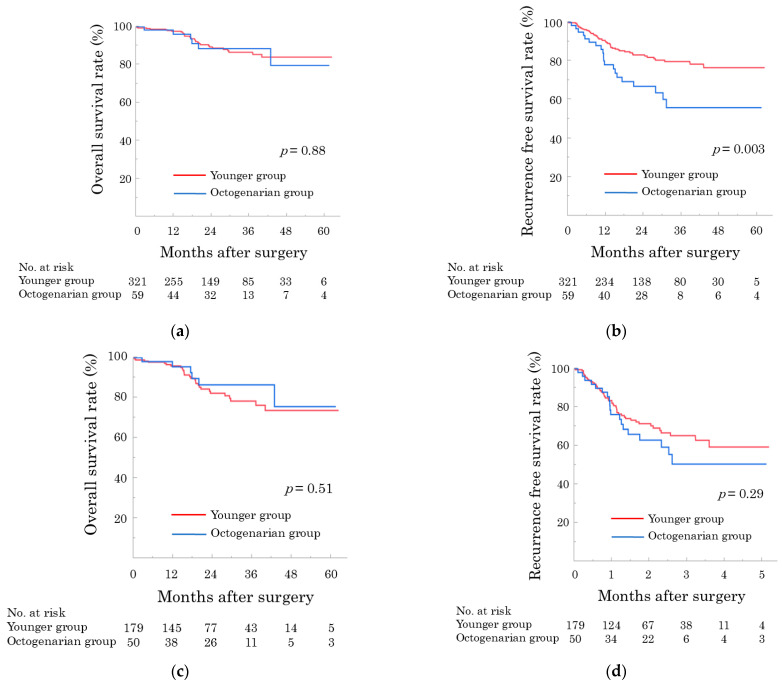
Survival after robotic pancreatectomy in younger and octogenarian patients. (**a**) Overall survival curves in all cohorts (*p* = 0.88). (**b**) Recurrence-free survival curves in all cohorts (*p* = 0.003). (**c**) Overall survival curves in patients with malignant diseases (*p* = 0.51) (**d**) Recurrence-free survival curves in patients with malignant diseases (*p* = 0.29).

**Table 1 cancers-17-03038-t001:** Baseline characteristics stratified by age.

Variables	n (%) or Median (IQR)	Younger Patients (<80 Years)	Octogenarian Patients (≥80 Years)	*p* Value
No. of patients	380	321	59	
Age, years	72 (61–77)	70 (59–75)	82 (81–85)	<0.001
Sex				
Men	196 (51.6%)	166 (51.7%)	30 (50.9%)	0.903
Women	184 (48.4%)	155 (48.3%)	29 (49.1%)	
BMI, kg/m^2^	22.4 (20.4–24.7)	22.3 (20.4–24.9)	22.7 (20.9–24.0)	0.970
ASA score				
1–2	325 (85.5%)	279 (86.9%)	46 (78.0%)	0.087
3–4	55 (14.5%)	42 (13.1%)	13 (22.0%)	
Comorbidity				
Hypertension	172 (45.3%)	138 (43.0%)	34 (57.6%)	0.038
Diabetes	118 (31.1%)	92 (28.7%)	26 (44.1%)	0.022
Neoadjuvant chemotherapy	72 (18.9%)	68 (21.2%)	4 (6.8%)	0.004
Primary diseases				
Pancreatic cancer	115 (30.3%)	92 (28.7%)	23 (39.0%)	<0.001
Bile duct cancer	30 (7.9%)	17 (5.3%)	13 (22.0%)	
IPMN/IPMC	83 (21.8%)	76 (23.7%)	7 (11.9%)	
PNEN	40 (10.5%)	37 (11.5%)	3 (5.1%)	
Ampullary carcinoma	24 (6.3%)	19 (5.9%)	5 (8.5%)	
Duodenal carcinoma	25 (6.6%)	20 (6.2%)	5 (8.5%)	
Others	63 (16.6%)	60 (18.7%)	3 (5.1%)	
Malignant diseases	229 (60.3%)	179 (55.8%)	50 (84.8%)	<0.001
Type of procedure				
RPD	213 (56.1%)	177 (55.1%)	36 (61.0%)	0.401
RDP	167 (43.9%)	144 (44.9%)	23 (39.0%)	
Institution				
Fujita Health University Hospital	199 (52.4%)	165 (51.4%)	34 (57.6%)	0.378
Okayama University Hospital	181 (47.6%)	156 (48.6%)	25 (42.4%)	

BMI, body mass index; ASA, American Society of Anesthesiologists; IPMN, intraductal papillary mucinous neoplasm; IPMC, intraductal papillary mucinous carcinoma; PNEN, pancreatic neuroendocrine neoplasm; RPD, robotic pancreatoduodenectomy; RDP, robotic distal pancreatectomy.

**Table 2 cancers-17-03038-t002:** Outcomes of robotic pancreatoduodenectomy stratified by age.

Variables	n (%) or Median (IQR)	Younger Patients (<80 Years)	Octogenarian Patients (≥80 Years)	*p* Value
No. of patients	213	177	36	
Preoperative biliary drainage	68 (31.9%)	48 (27.1)	20 (55.6)	0.001
*Operative factors*				
Operative time, min	490 (410–632)	478 (406–633)	516 (444–606)	0.256
Blood loss, mL	100 (38–242)	100 (30–228)	117 (72–338)	0.077
Conversion to open surgery, n (%)	3 (1.4%)	3 (1.7%)	0 (0%)	0.290
Transfusion, n (%)	17 (8.1%)	10 (5.7%)	7 (20.0%)	0.012
Pancreatic texture, n (%)				
Soft	170 (80.2%)	140 (79.6%)	30 (83.3%)	0.597
Hard	42 (19.8%)	36 (20.4%)	6 (16.7%)	
MPD diameter, mm	3 (2–4)	2.5 (2–4)	3 (2–4.8)	0.224
*Postoperative factors*				
Postoperative hospital stay, days	16 (11–23)	14 (10–22)	22 (16–27)	<0.001
30-day mortality, n (%)	2 (0.9%)	2 (1.1%)	0 (0%)	0.388
90-day mortality, n (%)	2 (0.9%)	2 (1.1%)	0 (0%)	0.388
Re-operation, n (%)	9 (5.0%)	7 (4.0%)	2 (5.6%)	0.675
Major complications, n (%)	37 (17.4%)	30 (17.0%)	7 (19.4%)	0.722
Incisional SSI, n (%)	9 (4.2%)	9 (5.1%)	0 (0%)	0.065
Organ/space SSI, n (%)	18 (8.5%)	14 (7.9%)	4 (11.1%)	0.543
POPF (≥grade B), n (%)	25 (11.7%)	21 (11.9%)	4 (11.1%)	0.898
Bile leakage, n (%)	5 (2.3%)	5 (2.8%)	0 (0%)	0.171
DGE (≥grade B), n (%)	11 (5.2%)	6 (3.4%)	5 (13.9%)	0.023
Postoperative pancreatitis	0 (0%)	0 (0%)	0 (0%)	-

IQR, interquartile range; MPD, main pancreatic duct; SSI, surgical site infection; POPF, postoperative pancreatic fistula; DGE, delayed gastric emptying.

**Table 3 cancers-17-03038-t003:** Outcomes of robotic distal pancreatectomy stratified by age.

Variables	n (%) or Median (IQR)	Younger Patients (<80 Years)	Octogenarian Patients (≥80 Years)	*p* Value
No. of patients	167	144	23	
*Operative factors*				
Type of procedure				
RDP with splenectomy	119 (71.3%)	100	19	0.207
Spleen-preserving technique	48 (28.7%)	44	4	
Kimura technique	25	24	1	
Warshaw technique	23	20	3	
Operative time, min	281 (219–372)	280 (217–372)	297 (238–398)	0.555
Blood loss, mL	50 (9–110)	35 (5–106)	58 (22–120)	0.144
Conversion to open surgery, n (%)	1 (0.6%)	1 (0.7%)	0 (0%)	0.586
Transfusion, n (%)	4 (2.4%)	3 (2.1%)	1 (4.3%)	0.523
*Postoperative factors*				
Postoperative hospital stay, days	11 (9–15)	10.5 (9–14.8)	14 (9–21)	0.023
30-day mortality, n (%)	0 (0%)	0 (0%)	0 (0%)	-
90-day mortality, n (%)	0 (0%)	0 (0%)	0 (0%)	-
Re-operation, n (%)	1 (0.6%)	0 (0%)	1 (4.3%)	0.045
Major complications, n (%)	13 (7.8%)	9 (6.3%)	4 (17.4%)	0.097
Incisional SSI, n (%)	6 (3.6%)	5 (3.5%)	1 (4.3%)	0.838
Organ/space SSI, n (%)	16 (9.6%)	12 (8.3%)	4 (17.4%)	0.205
POPF (≥grade B), n (%)	28 (16.8%)	23 (16.0%)	5 (21.7%)	0.504
Postoperative pancreatitis	2 (1.2%)	2 (1.4%)	0 (0%)	0.440

IQR, interquartile range; RDP, robotic distal pancreatectomy; SSI, surgical site infection; POPF, postoperative pancreatic fistula.

**Table 4 cancers-17-03038-t004:** Univariate and multivariable analyses of risk factors associated with major postoperative complications after robotic pancreatectomy.

Variables	Univariate Analysis			Multivariable Analysis		
	OR	95% CI	*p* Value	OR	95% CI	*p* Value
Age (≥80 years)	1.66	0.76–3.37	0.193	1.33	0.59–2.84	0.479
Sex (male)	2.73	1.45–5.41	0.002	2.53	1.31–5.10	0.005
BMI (≥25 kg/m^2^)	1.17	0.58–2.27	0.647			
ASA (3–4)	0.96	0.38–2.13	0.918			
Hypertension	1.03	0.57–1.88	0.911			
Neoadjuvant chemotherapy	0.24	0.06–0.69	0.005	0.26	0.06–0.77	0.012
Preoperative biliary drainage	1.17	0.53–2.40	0.681			
Etiology of disease						
Pancreatic cancer	1					
Bile duct cancer	3.86	1.15–12.6	0.030			
IPMN/IPMC	1.88	0.67–5.46	0.229			
PNEN	1.67	0.42–5.85	0.445			
Ampullary carcinoma	4.06	1.10–14.1	0.036			
Duodenal carcinoma	3.86	1.05–13.3	0.042			
Others	4.41	1.72–12.3	0.002			
Procedure (RPD)	2.49	1.31–5.02	0.005	2.02	1.04–4.16	0.038
Operative time (>500/280 min) *	0.98	0.54–1.78	0.949			
Blood loss (>500 mL)	2.80	1.11–6.52	0.031	2.27	0.85–5.62	0.099
Institution (Okayama)	1.11	0.61–2.03	0.719			

* longer than median operative time: >500 min for RPD and >280 for RDP. OR, odds ratio; CI, confidence interval; BMI, body mass index; ASA, American Society of Anesthesiologists; IPMN, intraductal papillary mucinous neoplasm; IPMC, intraductal papillary mucinous carcinoma; PNEN, pancreatic neuroendocrine neoplasm; RPD, robotic pancreatoduodenectomy.

## Data Availability

The raw data supporting the conclusions of this article will be made available by the authors on request.

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
