# Peer review of "Outcomes of Robotic Pancreatectomy in the Octogenarian: A Multicenter Retrospective Cohort Study"

_cancers, 2025, doi:10.3390/cancers17183038_

Round 1
Reviewer 1 Report
Comments and Suggestions for Authors
This is a valuable and timely study addressing an important clinical question regarding the feasibility of major robotic pancreatic surgery in the growing elderly population. The following suggestions are intended to further strengthen the manuscript.
1.
The manuscript's reliance on a chronological age cutoff (≥80 years) is a limitation, as this approach can be a crude surrogate for a patient's true physiological condition. To strengthen the analysis, the authors should consider incorporating a more sophisticated preoperative assessment of the patients' functional or "biological age" into the baseline comparison.
Instead of relying solely on comorbidities, including data on frailty scores, sarcopenia (which can be assessed from preoperative CT scans), or performance status (e.g., ECOG) would provide a much more meaningful comparison between the two cohorts. This would better reflect the patients' fitness for surgery and significantly enhance the clinical relevance of the study's findings, moving beyond an outdated, simple age-based stratification.
2. Inconsistent Patient Numbers : There is a critical discrepancy in the reported number of octogenarian patients. Please ensure the correct number is used consistently throughout the entire manuscript, including the Abstract, Simple Summary, and main text.
3. Error in Table 1 Data: In Table 1, the percentage for "Pancreatic cancer" in the "Younger patients" column is listed as 8.7%. Based on the patient number provided (n=92) out of a total of 321 younger patients, the correct percentage should be 28.7% (92/321).
4. Clarification of Multivariable Analysis Method: The rationale for selecting variables for the multivariable analysis in Table 4 is unclear.
5. Nuanced Interpretation of RDP Safety: The Discussion and Conclusion state that RDP is safe and feasible in octogenarians. While major complications were not statistically different, the manuscript should acknowledge the clinically concerning trends. The major complication rate was nearly three times higher (17.4% vs. 6.3%) and, as noted above, the re-operation rate was significantly higher in the octogenarian group. A more cautious interpretation is advisable, acknowledging that the small sample size (n=23) may be underpowered to detect a true difference in complication rates and that further study is needed.
6. Elaboration on "Careful Patient Selection": The manuscript repeatedly concludes that the procedure is safe for "carefully selected octogenarian patients". This is a key point that would be much more impactful if elaborated upon. Please provide details on the specific criteria used for this selection at your institutions
7. The figures could be improved for clarity. The text refers to "Figure 1A", but the figure itself is not labeled with "A". Furthermore, there are multiple charts presented under the "Figure 1" and "Figure 2" headings. Please review the figure numbering and ensure that all figures and sub-panels (e.g., 2a, 2b) are clearly labeled and referenced correctly in the text for ease of reading.
8. Discussion of Longer Hospital Stays: The authors plausibly attribute longer hospital stays in the elderly to baseline comorbidities. The discussion could be strengthened by exploring the direct postoperative causes.
Comments on the Quality of English Languageaccpetable but should be improved
Author Response
September 9, 2025
Maen Abdelrahim
Editor-in-Chief
Cancers
Dear Editor:
RE: cancers-3851556
Outcomes of robotic pancreatectomy in the octogenarian: A multicenter retrospective cohort study
Thank you for reviewing our manuscript titled “Outcomes of robotic pancreatectomy in the octogenarian: A multicenter retrospective cohort study”.
We are pleased that our manuscript was favorably reviewed and found to be potentially acceptable for publication pending revisions.
We thank the reviewer for his/her valuable insight and comments as these serve to further strengthen our manuscript.
As requested, we have provided a point-by-point response to the comments below with relevant changes made to the manuscript.
Reviewer #1
This is a valuable and timely study addressing an important clinical question regarding the feasibility of major robotic pancreatic surgery in the growing elderly population. The following suggestions are intended to further strengthen the manuscript.
Comment 1:
The manuscript's reliance on a chronological age cutoff (≥80 years) is a limitation, as this approach can be a crude surrogate for a patient's true physiological condition. To strengthen the analysis, the authors should consider incorporating a more sophisticated preoperative assessment of the patients' functional or "biological age" into the baseline comparison.
Instead of relying solely on comorbidities, including data on frailty scores, sarcopenia (which can be assessed from preoperative CT scans), or performance status (e.g., ECOG) would provide a much more meaningful comparison between the two cohorts. This would better reflect the patients' fitness for surgery and significantly enhance the clinical relevance of the study's findings, moving beyond an outdated, simple age-based stratification.
Response 1:
Thank you for your feedback. We believe that one of our novelties in this study would be the investigation regarding the effectiveness and safety of robotic pancreatectomy in octogenarian patients (≥80 years). As the reviewer 1 pointed out, data on frailty scores, sarcopenia would be interesting and should be investigated in future studies. In the revised manuscript, we have discussed this issue in the limitations (page 10, line 274-277).
Comment 2:
Inconsistent Patient Numbers: There is a critical discrepancy in the reported number of octogenarian patients. Please ensure the correct number is used consistently throughout the entire manuscript, including the Abstract, Simple Summary, and main text.
Response 2:
Thank you for your feedback. In the revised manuscript, we have carefully checked patient numbers throughout the entire manuscript.
Comment 3:
Error in Table 1 Data: In Table 1, the percentage for "Pancreatic cancer" in the "Younger patients" column is listed as 8.7%. Based on the patient number provided (n=92) out of a total of 321 younger patients, the correct percentage should be 28.7% (92/321).
Response 3:
Thank you for your feedback. In the revised manuscript, we have corrected the percentage as pointed out (Table 1).
Comment 4:
Clarification of Multivariable Analysis Method: The rationale for selecting variables for the multivariable analysis in Table 4 is unclear.
Response 4:
Thank you for your feedback.
Comment 5:
Response 5:
Thank you for your feedback. All preoperative and operative variables with P <0.01 in univariate analysis were included in multivariable analyses. In the revised manuscript, we have added the methods (page 3, line 126-127).
Comment 6:
Nuanced Interpretation of RDP Safety: The Discussion and Conclusion state that RDP is safe and feasible in octogenarians. While major complications were not statistically different, the manuscript should acknowledge the clinically concerning trends. The major complication rate was nearly three times higher (17.4% vs. 6.3%) and, as noted above, the re-operation rate was significantly higher in the octogenarian group. A more cautious interpretation is advisable, acknowledging that the small sample size (n=23) may be underpowered to detect a true difference in complication rates and that further study is needed.
Response 6:
Thank you for your feedback. In the revised manuscript, we have changed the descriptions regarding the outcomes of RDP in the results (page 7, line 200-202) and discussions (page 9, line 252-254).
Comment 7:
Elaboration on "Careful Patient Selection": The manuscript repeatedly concludes that the procedure is safe for "carefully selected octogenarian patients". This is a key point that would be much more impactful if elaborated upon. Please provide details on the specific criteria used for this selection at your institutions
Response 7:
Thank you for your feedback. Patients meeting the following criteria were eligible for surgery: capable of making decisions and expressing their wishes regarding medical procedures, and independent activity of daily living and mobility. In the revised manuscript, we have provided selection criteria for surgery (page 3, line 112-115).
Comment 8:
Discussion of Longer Hospital Stays: The authors plausibly attribute longer hospital stays in the elderly to baseline comorbidities. The discussion could be strengthened by exploring the direct postoperative causes.
Response 8:
Thank you for your feedback. Preoperative characteristics in octogenarian patients, such as higher ASA scores, more comorbidities, and a higher incidence of malignant disease, may have contributed to the prolonged hospital stays observed after surgery. Accordingly, longer hospital stays in octogenarian patients may be explained by the fact that they required longer times for gastrointestinal and functional recovery. In the revised manuscript, we have discussed this issue (page 9-10, line 255-258).
Comment 9:
Comments on the Quality of English Language
accpetable but should be improved
Response 9:
Thank you for your feedback. The manuscript has been checked by Editage (www.editage.jp) for English language editing.
Sincerely,
Kosei Takagi, MD, PhD
Department of Gastroenterological Surgery, Okayama University Graduate School of Medicine, Dentistry, and Pharmaceutical Sciences, 2-5-1 Shikata-cho, Kita-ku, Okayama 700-8558, Japan
Tel: +81-86-223-7151; Fax: +81-86-221-8775; E-mail: kotakagi15@gmail.com
Reviewer 2 Report
Comments and Suggestions for Authors
From a biostats and clinical epidemiology point of view, here are some comments for the Authors:
- the study has been very well planned and reported
- line 31, multivariable analyses have to be defined, in your case as binary logistic regression
- line 69 multicenter, retrospective, better to say multicenter, observational retrospective
- line 112, it would be very interesting, if these data would be available, to investigate the post-operative survival of these pts, expressed as OS and PFS
- at the light of my previous comment, can you confirm that RP has been both surgically and oncologically adequate?
- at the light of my previous comment, it would be useful to be given the median follow-up for the whole cohort, as well as stratified by age/RP
- table 1, the huge differences in histology, for pts under/over 80 yrs, is a critical confounder and this role needs to be underlined
- all around the manuscript, p-values should be reported with 3-sign digits (i.e. 0.842)
- line 173 rates of mortality, do you mean the surgical one at d30?
- table 4, can you confirm that the major complications were 37+13=50?
- table 4 Preoperative chemotherapy, better to say neoadjuvant
- I suggest to add a crosstable, histology vs RDP/RPD, since it seems a deep confounder too (we have to prove that no selection bias depended on this potential association)
- line 214 RPD outcomes in octogenarian, once again, you only covered the surgical outcomes, and not the oncological one; survival analyses are lacking
- mind that, when studying the introduction over the time course of RP, in survival modeling this covariate has a marked time-dependent behavior and needs to be treated by the proper Cox time-dependent model
Author Response
September 9, 2025
Maen Abdelrahim
Editor-in-Chief
Cancers
Dear Editor:
RE: cancers-3851556
Outcomes of robotic pancreatectomy in the octogenarian: A multicenter retrospective cohort study
Thank you for reviewing our manuscript titled “Outcomes of robotic pancreatectomy in the octogenarian: A multicenter retrospective cohort study”.
We are pleased that our manuscript was favorably reviewed and found to be potentially acceptable for publication pending revisions.
We thank the reviewer for his/her valuable insight and comments as these serve to further strengthen our manuscript.
As requested, we have provided a point-by-point response to the comments below with relevant changes made to the manuscript.
Reviewer #2
From a biostats and clinical epidemiology point of view, here are some comments for the Authors:
Comment 1:
the study has been very well planned and reported
Response 1:
Thank you for your positive feedback. We appreciate his/her comments.
Comment 2:
line 31, multivariable analyses have to be defined, in your case as binary logistic regression
Response 2:
Thank you for your feedback. In the revised manuscript, we have defined multivariable analyses, as suggested (page 1, line 35).
Comment 3:
line 69 multicenter, retrospective, better to say multicenter, observational retrospective
Response 3:
Thank you for your feedback. In the revised manuscript, we have described “multicenter, observational retrospective”, as suggested (page 2, line 73).
Comment 4:
line 112, it would be very interesting, if these data would be available, to investigate the post-operative survival of these pts, expressed as OS and PFS
Response 4:
Thank you for your feedback. In the revised manuscript, we have investigated postoperative survivals, including overall survival (OS) and recurrence-free survival (RFS), using the Kaplan–Meier method (page 3, line 127-130). Moreover, we have demonstrated long-term outcomes after robotic pancreatectomy (page 8, line 215-220; Figure 3).
Comment 5:
at the light of my previous comment, can you confirm that RP has been both surgically and oncologically adequate?
Response 5:
Thank you for your feedback. In the revised manuscript, we investigated long-term outcomes in all cohorts and patients with malignant diseases; however, the results lacked disease-specific long-term outcomes. Since the indication for robotic pancreatectomy in octogenarian patients was a higher incidence of malignant diseases, our results may support the surgical and oncological adequacy of robotic surgery for octogenarian patients. Future studies should investigate the disease-specific long-term outcomes in octogenarian patients to provide further evidence on the role of robotic pancreatectomy in this population. We have discussed this issue in the limitations (page 10, line 280-285).
Comment 6:
at the light of my previous comment, it would be useful to be given the median follow-up for the whole cohort, as well as stratified by age/RP
Response 6:
Thank you for your feedback. In the revised manuscript, we have provided the median follow-up for the whole cohort and demonstrated the Kaplan–Meier curves (page 8, line 215-220; Figure 3).
Comment 7:
table 1, the huge differences in histology, for pts under/over 80 yrs, is a critical confounder and this role needs to be underlined
Response 7:
Thank you for your feedback. We agree that these are our limitations. In the revised manuscript, we have underlined this issue in the limitations (page 10, line 272-274).
Comment 8:
all around the manuscript, p-values should be reported with 3-sign digits (i.e. 0.842)
Response 8:
Thank you for your feedback. In the revised manuscript, we have reported p-values with 3-sign digits.
Comment 9:
line 173 rates of mortality, do you mean the surgical one at d30?
Response 9:
Thank you for your feedback. We meant it as 30-/90-day mortality (page 7, line 191).
Comment 10:
table 4, can you confirm that the major complications were 37+13=50?
Response 10:
Thank you for your feedback. We have confirmed that the major complications were 37+13=50. In the revised manuscript, we have described them clearly (page 7, line 205-206).
Comment 11:
table 4 Preoperative chemotherapy, better to say neoadjuvant
Response 11:
Thank you for your feedback. In the revised manuscript, we have changed “preoperative” chemotherapy to “neoadjuvant” chemotherapy.
Comment 12:
I suggest to add a crosstable, histology vs RDP/RPD, since it seems a deep confounder too (we have to prove that no selection bias depended on this potential association)
Response 12:
Thank you for your feedback. We could add a crosstable, histology vs RDP/RPD, as below. However, there should be a huge discrepancy between the RDP and RPD due to the different indications of the surgical procedure. We don’t think that the histological differences between the groups would not a deep confounder in this study.
|
|
RDP (n = 167) |
RPD (n = 213) |
P value |
|
Pancreatic cancer |
68 |
47 |
<0.001 |
|
Bile duct cancer |
0 |
30 |
|
|
IPMN/IPMC |
36 |
47 |
|
|
PNEN |
26 |
15 |
|
|
Ampullary carcinoma |
0 |
24 |
|
|
Duodenal carcinoma |
0 |
25 |
|
|
Others |
37 |
25 |
|
Comment 13:
line 214 RPD outcomes in octogenarian, once again, you only covered the surgical outcomes, and not the oncological one; survival analyses are lacking
Response 13:
Thank you for your feedback. In the revised manuscript, we have investigated postoperative survivals, including overall survival (OS) and recurrence-free survival (RFS), using the Kaplan–Meier method (page 3, line 127-130). Moreover, we have demonstrated long-term outcomes after robotic pancreatectomy (page 8, line 215-220; Figure 3).
Comment 14:
mind that, when studying the introduction over the time course of RP, in survival modeling this covariate has a marked time-dependent behavior and needs to be treated by the proper Cox time-dependent model
Response 14:
Thank you for your feedback. Although this study investigated long-term outcomes in all cohorts and patients with malignant diseases, the results lacked disease-specific long-term outcomes. We believe that future studies should investigate the disease-specific long-term outcomes in octogenarian patients to provide further evidence on the role of robotic pancreatectomy in this population. In the revised manuscript, we have discussed this issue in the limitations (page 10, line 280-286).
Sincerely,
Kosei Takagi, MD, PhD
Department of Gastroenterological Surgery, Okayama University Graduate School of Medicine, Dentistry, and Pharmaceutical Sciences, 2-5-1 Shikata-cho, Kita-ku, Okayama 700-8558, Japan
Tel: +81-86-223-7151; Fax: +81-86-221-8775; E-mail: kotakagi15@gmail.com
Reviewer 3 Report
Comments and Suggestions for Authors
Dear Authors, I read your manuscript with great interest. I am a hepatobiliary surgeon using robotic-assisted techniques and found your cohort compelling with important findings. I offer a few editorial suggestions to improve the manuscript.
- "3.3 Outcomes of robotic pancreatectomy—probably better for pancreatoduodenectomy? I kindly suggest dividing the results into RPD and RDP.
- If possible, could you examine the characteristics of patients who died within 90 days? Is there any significant association with being discharged home?
- line 210 missing citation of meta analysis
- The discussion is insufficiently developed. Only eight articles are cited. Please expand the discussion by reviewing the literature on the exaggerated inflammatory response in older adults and by incorporating the concept and measures of frailty, which is common among octogenarians. In addition, compare your results with findings from open surgical series.
- Postoperative pancreatitis is a recently described entity. Did you include this complication in your study?
Author Response
September 9, 2025
Maen Abdelrahim
Editor-in-Chief
Cancers
Dear Editor:
RE: cancers-3851556
Outcomes of robotic pancreatectomy in the octogenarian: A multicenter retrospective cohort study
Thank you for reviewing our manuscript titled “Outcomes of robotic pancreatectomy in the octogenarian: A multicenter retrospective cohort study”.
We are pleased that our manuscript was favorably reviewed and found to be potentially acceptable for publication pending revisions.
We thank the reviewer for his/her valuable insight and comments as these serve to further strengthen our manuscript.
As requested, we have provided a point-by-point response to the comments below with relevant changes made to the manuscript.
Reviewer #3
Dear Authors, I read your manuscript with great interest. I am a hepatobiliary surgeon using robotic-assisted techniques and found your cohort compelling with important findings. I offer a few editorial suggestions to improve the manuscript.
Comment 1:
"3.3 Outcomes of robotic pancreatectomy—probably better for pancreatoduodenectomy? I kindly suggest dividing the results into RPD and RDP.
Response 1:
Thank you for your feedback. In the revised manuscript, we have divided the results into RPD (3.3.1) and RDP (3.3.2) (page 5-6, line 163-181).
Comment 2:
If possible, could you examine the characteristics of patients who died within 90 days? Is there any significant association with being discharged home?
Response 2:
Thank you for your feedback. Mortality cases were inpatient mortality, which were not associated with sur-gery-related complications or being discharged home (page 6, line 168-169).
Comment 3:
line 210 missing citation of meta analysis
Response 3:
Thank you for your feedback. In the revised manuscript, we have added the citation of meta analysis (page 9, line 244).
Comment 4:
The discussion is insufficiently developed. Only eight articles are cited. Please expand the discussion by reviewing the literature on the exaggerated inflammatory response in older adults and by incorporating the concept and measures of frailty, which is common among octogenarians. In addition, compare your results with findings from open surgical series.
Response 4:
Thank you for your feedback. In the revised manuscript, we have expanded the discussion according to the insightful reviewers’ comments by adding the citations as well as by incorporating the concept and measures of frailty (page 9-10, line 243-286).
Comment 5:
Postoperative pancreatitis is a recently described entity. Did you include this complication in your study?
Response 5:
Thank you for your feedback. In the revised manuscript, we have included the data of postoperative pancreatitis (page 3, line 91; Table 2 and 3).
Sincerely,
Kosei Takagi, MD, PhD
Department of Gastroenterological Surgery, Okayama University Graduate School of Medicine, Dentistry, and Pharmaceutical Sciences, 2-5-1 Shikata-cho, Kita-ku, Okayama 700-8558, Japan
Tel: +81-86-223-7151; Fax: +81-86-221-8775; E-mail: kotakagi15@gmail.com
Round 2
Reviewer 1 Report
Comments and Suggestions for Authors
This manuscript is revised well and appropriately.
Reviewer 2 Report
Comments and Suggestions for Authors
The Authors were able to solve all previous concerns, congrats!
Reviewer 3 Report
Comments and Suggestions for Authors
Dear Authors, the new version is much improved. You have addressed or incorporated all of my suggestions, and I have no further objections. I recommend it for publication. Kind regards.